# Exploring Triaging and Short-Term Outcomes of Early Invasive Strategy in Non-ST Segment Elevation Acute Coronary Syndrome: A Report from Japanese Multicenter Registry

**DOI:** 10.3390/jcm9041106

**Published:** 2020-04-13

**Authors:** Nobuhiro Ikemura, Yasuyuki Shiraishi, Mitsuaki Sawano, Ikuko Ueda, Yohei Numasawa, Shigetaka Noma, Masahiro Suzuki, Yukihiko Momiyama, Kentaro Hayashida, Shinsuke Yuasa, Hiroaki Miyata, Keiichi Fukuda, Shun Kohsaka

**Affiliations:** 1Department of Cardiology, Keio University School of Medicine, Tokyo 160-8582, Japan; ikemu0129@gmail.com (N.I.); yasshiraishi@keio.jp (Y.S.); mitsuakisawano@gmail.com (M.S.); iueda@a7.keio.jp (I.U.); k-hayashida@umin.ac.jp (K.H.); yuasa@keio.jp (S.Y.); kfukuda@a2.keio.jp (K.F.); 2Department of Cardiology, Japanese Red Cross Ashikaga Hospital, Tochigi Prefecture 326-0843, Japan; numasawa@cpnet.med.keio.ac.jp; 3Department of Cardiology, Saiseikai Utsunomiya Hospital, Tochigi Prefecture 321-0974, Japan; sigetaka_noma@saimiya.com; 4Department of Cardiology, National Hospital Organization, Saitama Hospital, Saitama, Prefecture 351-0102, Japan; suzuki.masahiro.yd@mail.hosp.go.jp; 5Department of Cardiology, National Hospital Organization, Tokyo Medical Center, Tokyo 152-8902, Japan; ymomiyamajp@gmail.com; 6Department of Health Policy and Management, Keio University School of Medicine, Tokyo 160-8582, Japan; h-m@keio.jp

**Keywords:** acute coronary syndrome, percutaneous coronary intervention, acute kidney injury, quality of care

## Abstract

This observational study aimed to examine the extent of early invasive strategy (EIS) utilization in patients with non-ST elevation acute coronary syndrome (NSTE-ACS) according to the National Cardiovascular Data Registry (NCDR) CathPCI risk score, and its association with clinical outcomes. Using a prospective multicenter Japanese registry, 2968 patients with NSTE-ACS undergoing percutaneous coronary intervention within 72 hours of hospital arrival were analyzed. Multivariable logistic regression analyses were performed to determine predictors of EIS utilization. Additionally, adverse outcomes were compared between patients treated with and without EIS. Overall, 82.1% of the cohort (*n* = 2436) were treated with EIS, and the median NCDR CathPCI risk score was 22 (interquartile range: 14–32) with an expected 0.3–0.6% in-hospital mortality. Advanced age, peripheral artery disease, chronic kidney disease or patients without elevation of cardiac biomarkers were less likely to be treated with EIS. EIS utilization was not associated with a risk of in-hospital mortality; yet, it was associated with an increased risk of acute kidney injury (AKI) (adjusted odds ratio: 1.42; 95% confidence interval: 1.02–2.01) regardless of patients’ in-hospital mortality risk. Broader use of EIS utilization comes at the cost of increased AKI development risk; thus, the pre-procedural risk-benefit profile of EIS should be reassessed appropriately in patients with lower mortality risk.

## 1. Introduction

Non-ST elevation acute coronary syndrome (NSTE-ACS) remains the most frequent clinical presentation of acute coronary heart disease. It is known to entail a broad clinical spectrum with large variation in the incidence of in-hospital mortality ranging from less than 1% to 40% [1]. Hence, when managing the patients presenting with NSTE-ACS, it is of utmost importance to appropriately estimate individual risk, with subsequent, and preferably tailored invasive treatment. Several randomized studies have shown a trend toward greater clinical benefit when implementing an early invasive strategy (EIS), particularly in higher risk patients [2]. Consequently, the current clinical practice guidelines recommend the use of risk scores to stratify patients’ initial mortality risk [2,3].

However, observational studies, including ours, have reported a marked discrepancy between the patients’ predicted risk and the use of EIS [4,5]. This discrepancy may have widened in recent years; a report from U.S. showed an unnatural increase in diagnostic and/or coding unstable angina (UA) and consequent percutaneous coronary intervention (PCI) with a decrease in an inappropriate PCI for stable ischemic heart disease after being authorized by appropriate use criteria [6]. There also is a rising concern that unintended adverse outcomes associated with EIS have the potential to modify the beneficial effect of this approach, particularly for low-risk patients with NSTE-ACS. Specifically, recent studies have emphasized a high incidence of acute kidney injury (AKI) after cardiac catheterization in patients with NSTE-ACS, but data on the risk of AKI are limited because of the exclusion of patients with moderate to severe renal insufficiency from randomized trials [7,8]. 

Herein, we aimed to examine the extent of EIS utilization in patients with NSTE-ACS according to their predicted mortality risk to explore clinical predictors associated with EIS utilization and to further evaluate the association between EIS utilization and observed procedure-related adverse outcomes using a large-scale, multicenter PCI registry.

## 2. Methods

### 2.1. Data Sources

We used data from patients enrolled in the Japan Cardiovascular Database-Keio interhospital cardiovascular studies (JCD-KiCS) registry. The rationale and design of the JCD-KiCS registry have been described previously [9]. In brief the JCD-KiCS is a contemporary, prospective, multicenter registry, which collects clinical variables and outcome data on consecutive patients who have undergone PCI, with dedicated clinical research coordinators allocated to each hospital. The JCD-KiCS registry comprise 15 network hospitals within the metropolitan Kanto area, which include mostly large tertiary care hospitals (>200 beds; *n* = 13), with a few mid-sized satellite hospitals (<200 beds; *n* = 2). Participating hospitals were instructed to document and register data from consecutive hospital visits for PCI using an internet-based data collection system. The data registered were reviewed for completeness and internal consistency. Quality assurance of the data was achieved through automatic system validation and reporting of data completeness, education and training by clinical research coordinators who were specifically trained for the present PCI registry. 

The study protocol was approved by the institutional review board committee at each participating hospital. All participants provided written informed consent. Before the launch of the JCD-KiCS registry, information on the objectives of the study and its social significance, as well as an abstract, were provided for clinical trial registration with the University Hospital Medical Information Network (UMIN 000004736).

### 2.2. Study Population and Risk Stratification

We included patients with NSTE-ACS who underwent PCI within 3 days of hospital arrival between October 2008 and April 2016 (*n* = 3124) in order to focus on patients that would benefit the most from revascularization in an acute setting. We excluded 156 (4.9%) patients with missing data that precluded the ability to calculate the NCDR CathPCI risk score. Finally, we analyzed 2968 patients with NSTE-ACS who underwent PCI (Figure 1).

Clinical variables and in-hospital outcomes of the JCD-KiCS registry was defined in accordance with the National Cardiovascular Data Registry (NCDR) CathPCI registry version 4.1 and were developed in collaboration with the American College of Cardiology, which gave us a unique opportunity to directly compare practice differences between the United States (U.S.) and Japan. [9]. NSTE-ACS was defined as the presentation of unstable angina (UA) or non-ST elevation myocardial infarction (NSTEMI) on admission. UA (*n* = 1611, 54.3%) was defined as patients present with the following three principal presentations: 1.) rest angina (occurring at rest and prolonged, usually >20 minutes); 2.) new-onset angina (within the past 2 months, of at least Canadian Cardiovascular Society Class III severity); or 3.) increasing angina (previously diagnosed angina that has become distinctly more frequent, longer in duration, or increased by 1 or more Canadian Cardiovascular Society classes to at least CCS III severity). NSTEMI (*n* = 1357, 45.7%) was defined by the presence of both criteria as follows: 1.) Cardiac biomarkers (creatinine kinase-myocardial band, Troponin T or I) exceed the upper limit of normal according to the individual hospital’s laboratory parameters with a clinical presentation, which is consistent or suggestive of ischemia. Electrocardiogram (ECG) changes and/or ischemic symptoms may or may not be present. 2.) Absence of ECG changes diagnostic of a STEMI (new or presumed new sustained ST-segment elevation at the J-point in two contiguous ECG leads with the cut-off points: ≥0.2 mV in men or ≥0.15 mV in women in leads V2–V3 and/or ≥0.1 mV in other leads). 

The NCDR CathPCI mortality risk score was calculated for each patient, and patients were further divided into quartile groups according to their predicted risk. In brief, the NCDR CathPCI risk score was originally designed to predict in-hospital mortality [10]. It has been previously validated in our registry, and it demonstrated good discrimination and calibration [11]. Components of the NCDR CathPCI risk score include age, cardiogenic shock, prior chronic heart failure, peripheral artery disease (PAD), chronic lung disease, estimated glomerular filtration rate (eGFR), New York Heart Association Classification (NYHA) IV and PCI status. These variables were assessed by interventional cardiologists at each site, except for eGFR, which was calculated based on serum creatinine concentrations using revised equations for the Japanese population [12]

### 2.3. Measured Outcomes

We measured the incidence of all-cause death, bleeding complications, stroke during the patient’s initial hospitalization for PCI and AKI after PCI procedure. We defined in-hospital mortality as the rate of all-cause death during hospitalization. The post-procedural creatinine value was defined as the highest value within 30 days after the index procedure. If more than 1 post-procedural creatinine level was measured, the highest value was used for determining AKI. [13]. Given the short length of stay for many procedures, particularly for unstable angina cases, and the fact that peak creatinine levels are often observed 3 to 5 days after contrast exposure [14], the JCD-KiCS registry could have estimated the true rate of AKI in the real-world setting. In this study, we defined AKI as an absolute (>0.5 mg/dl) or relative (>25%) increase from baseline serum creatinine levels after the index procedure [15,16]. In addition, AKI requiring dialysis was identified by using a pre-defined JCD-KiCS data element for acute or worsening renal failure necessitating new renal dialysis after PCI [13]. We defined bleeding complications as those requiring a blood transfusion, and/or prolonging the hospital stay, and/or causing a decrease in the hemoglobin level >3.0 g/dL [17]. Periprocedural stroke was defined as a loss of neurological function caused by an ischemic or hemorrhagic event with residual symptoms lasting at least 24 hours after onset [18]. Length of stay was calculated as the total number of days from admission to discharge from a registered hospital.

### 2.4. Statistical Analysis

Baseline characteristics, procedural findings, relevant clinical outcomes such as in-hospital mortality, acute kidney injury, acute kidney injury requiring dialysis, bleeding complication within 72 h after PCI, stroke and length of stay were compared between patients treated with EIS (i.e., PCI performed within 24 hours after an administration) and delayed PCI (i.e., PCI performed after 24 to 72 after an administration). Continuous variables are presented as median and interquartile range (IQR), and categorical variables are presented as numbers and percentages. Group differences were evaluated using the chi-square test for categorical variables and the Wilcoxon rank-sum test for continuous variables.

Multivariable logistic regression analyses were performed after estimating the adjusted odds ratios (ORs) and 95% confidence intervals (CIs) for EIS utilization across NCDR CathPCI risk score quartiles, and adjusting for the following clinically relevant variables associated with utilization of EIS [4]: sex, age (≥75 years), prior heart failure, diabetes mellitus, PAD, chronic kidney disease (CKD; eGFR < 60 mL/min/1.73 m^2^), decompensated heart failure at arrival (NYHA class III or IV), increased cardiac troponin levels, prior revascularization (PCI and/or coronary artery bypass graft) and cardiogenic shock or cardiopulmonary arrest during the course. In addition, we constructed multivariable logistic regression models to estimate the effect of EIS utilization on in-hospital mortality, bleeding complications and AKI adjusting for the aforementioned, clinically relevant variables. Additionally, participating hospitals were included as a random effect to account for clustering of patients by site in these models. There were missing data for less than 2% of all candidate variables. The model for AKI was also applied to the following subgroups: sex, age (<75 or ≥75 years old), type of ACS (i.e., NSTEMI or UA), NCDR Cath PCI risk score (i.e., below or above the calculated median NCDR CathPCI risk score), CKD (eGFR <60 or ≥60 mL/min/1.73 m^2^), access site (radial or femoral) and contrast volume (above or below the median). All reported *p*-values were two-sided, and a *p*-value less than 0.05 was considered statistically significant. All analyses were performed using SPSS version 23.0 (IBM Corp., Armonk, NY, USA).

## 3. Results

Overall, the majority of the consecutively registered patients (82.1%, *n* = 2436) were treated with EIS. The calculated median NCDR CathPCI risk score was 22 (IQR, 14–32; Appendix A), equivalent to expected in-hospital mortality of 0.3–0.6% [9]. A proportional increase in EIS utilization was observed as patients’ risks increased according to the NCDR CathPCI risk score quartiles, with rates of 79.0% in the first quartile (Q1), 83.1% in the second quartile (Q2), 80.1% in the third quartile (Q3) and 87.3% in the fourth quartile (Q4; *p* < 0.001 for trend; Figure 2).

### 3.1. Baseline Characteristics of Patients Treated with and without EIS

Table 1 summarizes the baseline and procedural characteristics according to treatment with EIS and a delayed PCI. Patients with an increased cardiac troponin level (39.4% of patients treated with EIS versus vs. 15.8% of patients treated with a delayed PCI, *p* < 0.001) were more likely to be treated with EIS. Additionally, patients who experienced cardiogenic shock (3.1% vs. 1.5%, respectively, *p* = 0.046) or cardiopulmonary arrest (2.3% vs. 0.2%, respectively, *p* < 0.001) were more likely to be treated with EIS. In contrast, patients with older age, PAD, CKD, prior myocardial infarction and prior revascularization were less likely to be treated with EIS (*p* < 0.05 for all). Among the patients with available data for contrast volume (27%, 803/2968), patients with EIS were more likely to be treated with contrast than those with a delayed PCI (160 (IQR, 125–200) vs. 150 (IQR, 119.7–200), *p* = 0.007). Importantly, patients with EIS had a significantly higher NCDR CathPCI risk score than those with a delayed PCI (*p* < 0.01). Additionally, the baseline and procedural characteristics across patients with and without development of AKI are described in the Appendix A.

### 3.2. Predictors of EIS Utilization

The results of a multivariable analysis for predictors associated with EIS utilization are shown in Figure 3. A higher NCDR CathPCI risk score remained an independent predictor of EIS utilization: Q4, OR 3.56 (95% CI 2.34–5.43); Q3, OR 2.06 (95% CI 1.47–2.87); and Q2, OR 1.48 (95% CI 1.13–1.95), with Q1 used as a reference group. In addition, an increased cardiac troponin level was an independent predictor of EIS utilization (adjusted OR 3.42 (95% CI 2.63–4.54)). In contrast, patients with a higher age (adjusted OR 0.57 (95% CI 0.44–0.73)), PAD (adjusted OR 0.57 (95% CI 0.38–0.84)) or CKD (adjusted OR 0.60 (95% CI 0.45–0.78)) remained less likely to be treated with EIS, regardless of their predicted mortality risks.

### 3.3. In-Hospital Outcomes

There were no significant differences in rates of crude in-hospital mortality, bleeding complications, and stroke between patients treated with EIS and a delayed PCI (Table 2). However, patients treated with EIS were more likely to develop AKI (defined from maximum value of creatine within 30 days after the procedure) than those treated with a delayed PCI (15.0% vs. 9.9%, *p* = 0.004; Table 2). The incidence of AKI requiring dialysis was not statistically significant between two groups (0.9% vs. 1.1%, *p* = 0.69). Importantly, patients treated with EIS had a significantly shorter length of stay than those treated with a delayed PCI (median, 3.5 days (IQR 1.9–8.1) vs. 4.4 days (IQR 3.5–7.5), *p* < 0.001; Table 2).

After adjustments for known predictors, EIS utilization was independently associated with the development of AKI (adjusted OR 1.42 (95% CI 1.02–2.01), *p* = 0.04, Table 3), but it was not associated with in-hospital mortality (adjusted OR 0.99 (95% CI 0.43–2.29), *p* = 0.98), and bleeding complications (adjusted OR 0.91 (0.52–1.57), *p* = 0.72). Detailed results of the regression analysis are described in Appendix A. The subgroup analyses, representing the OR of EIS for developing AKI, showed that EIS utilization was consistently associated with higher likelihood of AKI (Figure 4). Importantly, the association between EIS utilization and a development of AKI was not significant in CKD patients (adjusted OR 0.92, 95% CI 0.57–1.46); on the contrary, the association was significant in non-CKD patients (adjusted OR 2.28, 95% CI 1.31–3.97). Furthermore, patients treated with a femoral approach were more likely to develop AKI (OR 1.53, 95% CI 1.01–2.34), albeit the trend was not statistically significant among those with a radial approach.

## 4. Discussion

In this present study, the rate of EIS utilization was approximately 80% in patients with NSTE-ACS, regardless of their individual mortality risk. Specifically, an increased cardiac troponin level was independently associated with EIS utilization, whereas a higher age, PAD or CKD were inversely associated with EIS utilization. In terms of procedure-related outcomes, EIS utilization was not associated with a risk of in-hospital mortality and bleeding complications; yet, it was associated with an increased risk of AKI.

Several large-scale trials have explored the optimal timing of invasive coronary angiography and revascularization to further improve clinical outcome for patients with NSTE-ACS [19,20]. Previously, Mehta et al. showed that EIS was more effective in reducing ischemic complications than a delayed invasive approach among patients at a high-risk (defined by a GRACE score > 140) [19]. On the basis of the aforementioned findings, clinical practice guidelines for NSTE-ACS management incorporated the use of risk scores to stratify mortality risk and EIS for patients with at least one high-risk criterion (abnormal cardiac troponin compatible with myocardial infarction, dynamic ECG changes, or a GRACE risk score > 140) since 2007 [21]. Consequently, the U.S. national database showed that clinical outcomes in patients with NSTE-ACS have progressively improved within the last two decades with an increasing EIS utilization (from 7.0% in 1995 to 36% in 2015), although a trend towards a smaller improvement in recent years [22]. The current clinical trial, very early vs. deferred invasive evaluation using computerized tomography (VERDICT), did not support an advantage of very early invasive strategy (i.e., cardiac catheterization within less than 12 hours) in all-comer patients compared with a more delayed invasive approach [20]. An important contemporary challenge in the management of patients with NSTE-ACS is to define the patients who are more likely to achieve a beneficial effect of EIS.

In our study, EIS utilization was independently associated with the development of AKI, despite patients with CKD being less likely to be treated with EIS. Patients without increases in serum creatinine level might have less chance of taking a sufficient rehydration therapy before PCI. Data on the risk of AKI for NSTE-ACS patients are limited because of the exclusion of patients with moderate to severe renal insufficiency from majority of the randomized trials [7,8]. Among clinically ill patients, the development of AKI accounts for 20–25% of the in-hospital mortality rate [23]; furthermore, survivors of AKI, even in patients with lower grade AKI, have an approximately sesquialteral higher adjusted risk of death than those without AKI [24]. Although our study did show shorter lengths of stay with EIS, we previously reported that the total amount of additive costs is known to be exceptionally high in the cases complicated with post-procedural AKI [25]. Together, these results indicate the importance of risk prediction for AKI and short-term mortality, and suggest that the incidence of AKI might modify the beneficial effect of EIS utilization on long-term mortality. Discrepancies between treatment efficacy and effectiveness may arise when a selectively beneficial, but potentially harmful, treatment is inappropriately applied to patients who are least likely to benefit.

Importantly, a substantial number of lower-risk patients were treated with EIS in our study. An increased cardiac troponin level is a high-risk characteristic that mandates an invasive approach in the most recent treatment guidelines [2]; in fact, it was a positive independent predictor of EIS utilization herein. There are sizable number of patients diagnosed as NSTEMI with elevation of creatinine kinase-myocardial band. The treating physicians might focus on the increased cardiac troponin level and fail to properly integrate low-risk or intermediate-risk patients with NSTE-ACS into an overall risk stratification. In addition, clinicians need to be vigilant since elevated troponin might be due to other causes than acute myocardial infarction such as chronic coronary or hypertensive heart disease [2]. Furthermore, patients with end-stage renal disease and no clinical evidence of ACS frequently have increased cardiac troponin levels [26]. In contrast, older age (≥75 years), PAD and CKD were negative independent predictors of EIS utilization, regardless of the risk score. Older age is accompanied by comorbidities and physiological frailty that adversely impact upon clinical outcomes. Although previous studies showed that the highest risk reduction in death/myocardial infarction with an EIS occurred in those more than 75 years of age, EIS was known to increase the risk for procedure-related complications, which may have affected the treating physician’s decision-making. Moreover, another study showed that age is the strongest risk factor for not undergoing EIS; therefore, our results were consistent with previous reports.

Of note, the expected in-hospital mortality rates, according to the NCDR CathPCI risk score, were relatively low in our study. We have previously shown a lower prevalence of predisposing cardiac risk factors and comorbidities in Japanese patients compared to that in U.S. patients [9]. However, nearly 80% of the acute procedures were categorized as appropriate under appropriate use criteria (AUC) 2009 and AUC 2012 [27]. It is possible that Japanese patients develop NSTE-ACS with a mild severity level instinctively, and it remains unclear whether Westernized risk stratification is suitable for Japanese patients. Current Japanese practice guidelines recommend a similar risk stratification as that in Western countries; however, based on our results, there appears to be a substantial need for further research on the indication of EIS for relatively low-risk patients.

The present study has several limitations. First, it has the inherent limitations of any nonrandomized observational research study, and unmeasured confounders among the associated variables may have contributed to the results, even after the rigorous measurement and recording of key clinical variables. As aforementioned, our registry was developed in accordance with NCDR version 4.1, which did not record non-quantifiable parameters (e.g., the presence of dementia or physical frailty), out-of-hospital variables or treatment within emergency medical systems. Second, consecutive patient enrollment was encouraged but could not be ascertained. Indeed, the relatively low in-hospital mortality rate implies that patients with early deaths were excluded. Although this may have introduced an unmeasurable selection bias, the findings should remain generalizable to survivors for whom risk stratification and EIS utilization are applicable. Third, the NCDR CathPCI risk score is objective and validated for predicting in-hospital mortality, but it includes procedure-related variables, which renders it difficult to pre-procedurally identify patients at risk of in-hospital mortality. Fourth, since the registry was aimed at evaluating patients who underwent PCI, we were unable to analyze the patients with NSTE-ACS who did not require any revascularization (i.e., underwent coronary angiography without PCI or treatment with medical management only). However, data from the coronary care unit network of the Tokyo metropolitan area showed that most (87.7%) patients with NSTE-ACS underwent coronary angiography, and the proportion of those who did not require any revascularization was relatively low (5.9%) [28]. Lastly, we analyzed treatment only at hospitals with revascularization capabilities, which may lead to the overestimation of early invasive management and consequent procedure-related outcomes.

## 5. Conclusions

EIS was utilized in most patients with NSTE-ACS regardless of their in-hospital mortality risk in the contemporary Japanese registry. EIS utilization is associated with an increased risk of AKI development; thus, the risk-benefit profile of EIS in patients with lower mortality risk should be reassessed.

## Figures and Tables

**Figure 1 jcm-09-01106-f001:**
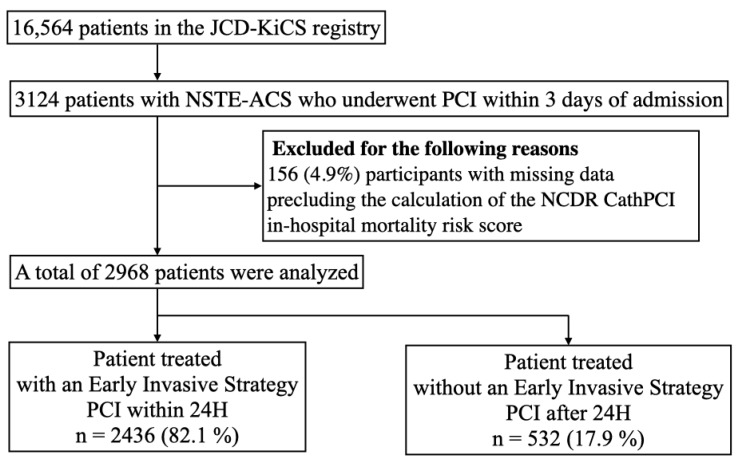
Study flowchart. Abbreviations: JCD-KiCS, Japan Cardiovascular Database-Keio interhospital cardiovascular studies; NSTE-ACS, non-ST elevation acute coronary syndrome; PCI, percutaneous coronary intervention; NCDR, national cardiovascular data registry; H, hour.

**Figure 2 jcm-09-01106-f002:**
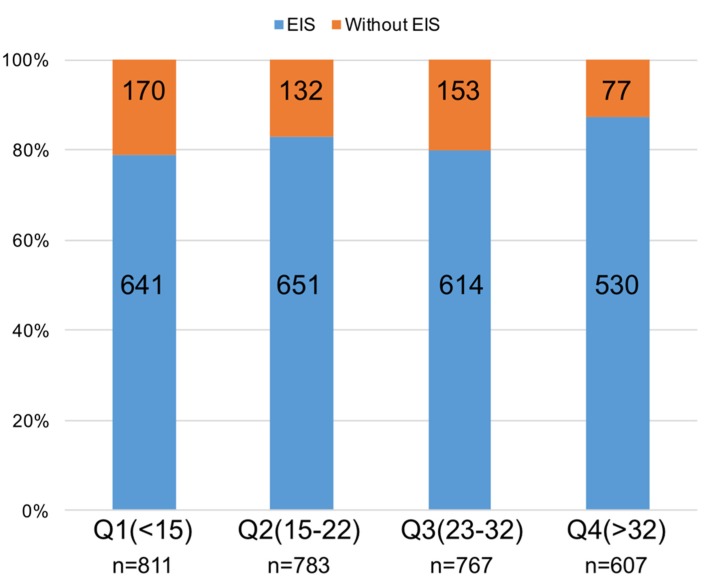
Percentages of patients treated with an early invasive strategy, according to NCDR CathPCI risk score quartiles. Abbreviations: EIS, early invasive strategy; NCDR, national cardiovascular data registry.

**Figure 3 jcm-09-01106-f003:**
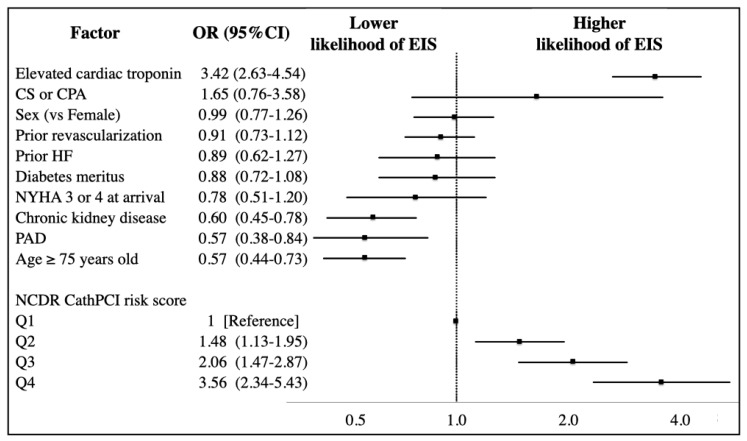
Important independent predictors of early invasive strategy utilization. Adjusted ORs (point estimate) and 95% CIs (error bars) indicate the likelihood of early invasive strategy utilization. ORs < 1 indicate decreased odds of early invasive strategy utilization. Covariates: sex, age (≥75 years), prior heart failure, diabetes mellitus, PAD, chronic kidney disease (eGFR < 60 mL/min/1.73 m^2^), decompensated heart failure at arrival (NYHA class III or IV), increased cardiac troponin levels, prior revascularization (PCI and/or coronary artery bypass graft), cardiogenic shock or cardiopulmonary arrest during the course. Abbreviations: CI, confidence interval; CKD, chronic kidney; CPA, cardiopulmonary arrest; disease; CS, cardiogenic shock; EIS, early invasive strategy; GFR, glomerular filtration rate; HF, heart failure; NYHA, New York Heart Association; PAD, peripheral arterial disease; OR, odds ratio.

**Figure 4 jcm-09-01106-f004:**
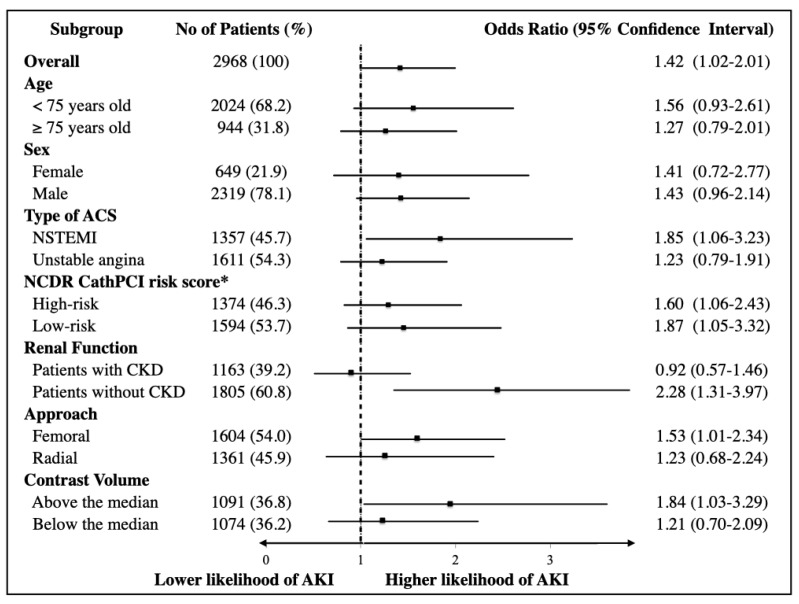
The association between EIS utilization and acute kidney injury. Subgroup results for adjusted ORs (point estimate) and 95% CIs (error bars), which indicate the likelihood of acute kidney injury with an early invasive strategy utilization. ORs > 1 indicate increased odds of acute kidney injury with an early invasive strategy utilization. All analyses were adjusted for sex, age (≥75 years), prior heart failure, diabetes mellitus, PAD, chronic kidney disease (eGFR < 60 mL/min/1.73 m^2^), decompensated heart failure at arrival (NYHA class III or IV), increased cardiac troponin levels, prior revascularization (PCI and/or coronary artery bypass graft) and cardiogenic shock or cardiopulmonary arrest during the course. *Below and above the calculated median NCDR CathPCI risk score. Abbreviations; ACS, acute coronary syndrome; NSTEMI, non-ST elevation myocardial infarction; early invasive strategy; NCDR, National Cardiovascular Data Registry; CKD, chronic kidney disease; AKI, acute kidney injury

**Table 1 jcm-09-01106-t001:** Baseline characteristics.

Number (%)	Treated with an EIS (PCI ≤ 24 h; *n* = 2436)	Treated with Delayed PCI (PCI 24–72 h; *n* = 532)	*p* Value
Male	1907 (78.3)	412 (77.4)	0.67
Age, median (Q1–Q3)	69 (60–76)	71 (63–78)	<0.001
BMI, median (Q1–Q3)	23.7 (21.5–25.9)	23.8 (22.0–26.1)	0.83
Family history of CAD	299 (12.3)	66 (12.5)	0.93
Current smoker	856 (35.2)	165 (31.0)	0.069
Past medical history			
Hypertension	1836 (75.4)	416 (78.2)	0.16
Diabetes mellitus	945 (38.8)	223 (41.9)	0.18
Dyslipidemia	1610 (66.1)	359 (67.5)	0.53
CKD	928 (38.1)	235 (44.2)	0.009
eGFR, median (Q1–Q3)	69.0 (48.5–92.9)	65.4 (43.9–82.1)	<0.001
HD	127 (5.2)	31 (5.8)	0.56
PAD	151 (6.2)	47 (8.8)	0.027
COPD	81 (3.3)	17 (3.2)	0.88
Prior heart failure	152 (6.2)	35 (6.6)	0.77
Prior MI	433 (17.8)	127 (23.9)	0.001
Prior PCI/CABG	725 (29.8)	198 (37.2)	0.001
Situation at arrival			
NSTEMI	1201 (51.3)	156 (33.3)	<0.001
Unstable angina	1235 (50.7)	376 (70.7)	<0.001
Heart failure	268 (11.0)	63 (11.8)	0.57
NYHA 3 or 4	159 (6.5)	34 (6.4)	0.90
EF, median (Q1–Q3)	60 (50–68)	61 (53.2–69)	0.77
EF <40%	121 (10.1)	24 (7.4)	0.13
Cardiogenic shock	75 (3.1)	8 (1.5)	0.046
CPA	57 (2.3)	1 (0.2)	0.001
Elevated cardiac troponin	922 (39.4)	74 (15.8)	<0.001
NCDR CathPCI risk score, median (Q1–Q3)	22 (14–32)	22 (12–31)	0.002
NCDR CathPCI risk score, mean, (SD)	24.3 (14.1)	21.9 (12.7)	<0.001
Procedure characteristics			
Femoral approach	1321 (54.2)	283 (53.2)	0.77
Use of IABP	169 (6.9)	28 (5.3)	0.15
Use of VA-ECMO	21 (0.9)	1 (0.2)	0.10
LMT lesion	96 (3.9)	25 (4.7)	0.42
LAD lesion	1242 (51.0)	264 (49.6)	0.56
LCX lesion	656 (26.9)	130 (24.4)	0.23
RCA lesion	716 (29.4)	171 (32.1)	0.21
Multivessel PCI	267 (11.0)	54 (10.2)	0.58
Fluoroscopy time, min, median, (Q1–Q3)	23.0 (16.0–35.1)	23.0 (15.3–35.1)	0.82
Contrast volume, mL median, (Q1–Q3)	160 (125–200)	150 (119.7–200)	0.007

Abbreviations: EIS, early invasive strategy; PCI, percutaneous coronary intervention; BMI, body mass index; CAD, coronary arterial disease; CKD, chronic kidney disease (eGFR < 60 mL/min/1.73 m^2^); eGFR, estimated glomerular filtration rate; HD, hemodialysis; MI, myocardial infarction; PAD, peripheral artery disease; COPD, chronic obstructive pulmonary disease; CABG, coronary artery bypass grafting; NSTEMI, non-ST elevation myocardial infarction; NYHA, New York Heart Association classification; EF, ejection fraction; CPA, cardiopulmonary arrest; NCDR, National Cardiovascular Data Registry; IABP, intra-aortic balloon pump; VA-ECMO, veno-arterial extracorporeal membrane oxygenation; LMT, left main trunk; LAD, left anterior descending; LCX, left circumflex; RCA, right coronary artery; Q, quartile.

**Table 2 jcm-09-01106-t002:** In-hospital outcomes.

Number (%)	Treated with an EIS (PCI ≤ 24 h; *n* = 2436)	Treated with a Delayed PCI (PCI 24–72 h; *n* = 532)	*p* Value
In-hospital mortality	50 (2.1)	7 (1.3)	0.26
Acute kidney injury*	329 (15.0)	46 (9.9)	0.004
Acute kidney injury requiring dialysis	23 (0.9)	6 (1.1)	0.69
Bleeding complication within 72 h	85 (3.5)	16 (3.0)	0.57
Stroke	9 (0.4)	1 (0.2)	0.51
Length of stay, days (IQR)	3.5 (1.9–8.1)	4.4 (3.5–7.5)	<0.001

*Acute kidney injury was defined as an absolute (>0.5 mg/dl) or relative (>25%) increase from baseline serum creatinine levels within 30 days of hospitalization. Abbreviations: EIS, early invasive strategy; PCI, percutaneous coronary intervention; IQR, inter quartile range.

**Table 3 jcm-09-01106-t003:** Factors associated with development of acute kidney injury.

Variables	Adjusted OR	95% Confidence Interval		*p* Value
		Lower Limit	Upper Limit	
Early invasive strategy	1.43	1.02	2.01	0.04
Male (vs. female)	0.71	0.55	0.92	0.011
Age (≥75 years old)	2.00	1.48	2.70	<0.001
Prior heart failure	1.80	1.27	2.55	0.001
Diabetes mellitus	1.32	1.05	1.67	0.018
Peripheral artery disease	0.75	0.46	1.23	0.25
Chronic kidney disease*	0.84	0.62	1.13	0.25
NYHA 3 or 4	2.09	1.44	3.05	<0.001
Elevated cardiac troponin	1.73	1.37	2.18	<0.001
Prior PCI/CABG	0.63	0.48	0.84	0.001
Cardiogenic shock/CPA	2.29	1.34	3.91	0.002

* Estimated glomerular filtration rate <60 mL/min/1.73 m^2^. Abbreviations: OR, odds ratio; NYHA, New York Heart Association classification; PCI, percutaneous coronary intervention; CABG, coronary artery bypass grafting; CPA, cardiopulmonary arrest.

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
