# Peer review of "Exploring Triaging and Short-Term Outcomes of Early Invasive Strategy in Non-ST Segment Elevation Acute Coronary Syndrome: A Report from Japanese Multicenter Registry"

_jcm, 2020, doi:10.3390/jcm9041106_

Round 1

Reviewer 1 Report

In the manuscript  “Exploring triaging and short-term outcomes of early invasive strategy in non-ST segment elevation acute coronary syndrome: A report from Japanese multicenter registry” by Ikemura et al. the authors describe their findings from a prospective registry for the use of an early invasive strategy (EIS) in patients with non-STE ACS. They found a high percentage of EIS use, without differences in mortality, stroke or bleedings. However acute kidney injury was more frequent in patients treated with EIS.

The analyses are appropriate, and the manuscript is well written. Although, there are a lot of reports on the potential benefits and harms of early invasive strategies, less is known on procedural outcomes and risks in non-Western ethnicities, so these findings are of scientific interest.

Some points should be considered before publication.

Please specify the used troponin assay, and the used threshold level for the diagnosis of NSTEMI. Was the diagnosis based on national guidelines or the universal definition? If the latter, which version? Please clarify.

Please provide, time of symptoms onset until presentation (since this would probably have an effect on the used strategy) contrast volume and procedural time in the baseline characteristics.

The dynamic changes of troponin over time, would be of interest as well. Patients with a high troponin delta between two measurements might profit most from EIS, since they probably have on-going acute ischemia. Maybe you can give a subgroup analysis of patients with high troponin deltas.

An unanswered question is, if you mainly observed acute kidney injury on top of already known chronic kidney injury? Or are these mainly new onset kidney injuries. Therefore, although not prespecified, please add CKD, contrast volume and access site (femoral/radial) to the cox regression model shown in figure S2. I would further recommend to show Table S2 in the main manuscript, since the development of AKI is your main finding.

Relating to this, please provide a baseline table in the supplement, stratifying patients by acute kidney injury. This could give an estimate on which predictors might further affect AKI irrespective of the used invasive strategy.

Where there any cases of patients requiring renal replacement therapy after PCI? If yes was this different between EIS and delayed PCI?

You state that AKI is a major contributor to additive costs in PCI. However, the referenced analysis from your group estimated 9.840 US $ additional costs per patient, if they develop AKI. I read your previous paper, and I do not fully understand why AKI does add such a high number. If I understood correctly, the main driver was length of stay. Naturally, this is affected by periprocedural AKI, however to see these high additional costs length of stay needs to increase tremendously or a high amount of additional expensive therapies (e.g. renal replacement) would be needed. Please explain a bit further in the discussion.

In line 269 you state: “Solitary increases of the troponin level cannot be assumed to be due to myocardial infarction because they can be due to other cardiac disorders.”  This sentence makes no sense to me. I see were you are aiming with this but please rephrase.

Reviewer 2 Report

Dear Editor

It is a great honour to review the manuscript “Exploring triaging and short-term outcomes of early 2 invasive strategy in non-ST segment elevation acute 3 coronary syndrome: A report from Japanese 4 multicentre registry” by Ikemura et al. All in all it is a well written manuscript that tackles some important questions.

Please find my comments below.

Abstract:

Line 31: No comma after EIS, and

Line 35: No a after increased

Line 36: Patients severity of what?

Introduction:

No comments on the introduction. It is well written.

Methods:

Please provide a literature reference for the definition of NSTE-ACS.

Statistical analysis;

What do you mean with relevant clinical outcomes? Please define.

Results:

Ample patient size.

In the baseline characteristics there is a discrepancy between the amount of patients with NSTEMI (1201pts and 156pts) and with elevated troponin (922pts and 74pts). Could you explain this? Were other biomarkers used (CK-CKMB?). This should also be addressed in the discussion section as Troponin is an independent predictor for EIS.

Also the rate of femoral access is quite high in both arms. Although I must acknowledge that radial access was not completely accepted it is still quite high. This could of course influence AKI. Please explain.

Discussion:

Page 8. Line 244 Current clinical trial etc. Please rephrase this sentence. Do you mean the current study or the VERDICT trial? It is not completely clear what you mean.

Page 8 line 247: The word patient should be plural; patients

There are some crooked sentences that might be avoided with the help of a native speaker.

Round 2

Reviewer 1 Report

I have no major comments to add.

However, I am still not happy with the phrasing of "Increases of the troponin level cannot be assumed to be due to myocardial infarction because it can occur in other cardiac disorders such as chronic coronary or hypertensive heart disease." in the discussion section.

Please change to: "In addition, clinician need to be vigilant since elevated troponin might be due to other causes than acute myocardial infarction such as chronic coronary or hypertensive heart disease."

or something like that.
